# Design and Analysis of Quartz Crystal Microbalance with a New Ring-Shaped Interdigital Electrode

**DOI:** 10.3390/s22197422

**Published:** 2022-09-29

**Authors:** Pengyi Wang, Mingxiang Ling, Minghai Li

**Affiliations:** 1Institute of Systems Engineering, China Academy of Engineering Physics, Mianyang 621999, China; 2Institute of Smart City and Intelligent Transportation, Southwest Jiaotong University, Chengdu 610031, China

**Keywords:** quartz crystal microbalance, interdigital electrode, uniform mass sensitivity, high quality factor

## Abstract

In this paper, a new type of ring-shaped interdigital electrode is proposed to improve the accuracy and repeatability of quartz crystal microbalance. The influence of different types of single finger, dot finger, dot double-finger electrodes on mass sensitivity distribution as well as the optimal proportion of finger and gap width are obtained through multi-physical coupling simulation. The results show that the design criteria of interdigital electrodes will not change with the increase in the number of fingers. The gap width should obey the decrease order from central to edge and be about twice the width of finger. The width of the outermost finger and the radius of the middle dot electrode should be maintained at about 0.4 and 0.2 times of the total electrode radius. An experiment was carried out to verify that the quartz wafer with a dot double-finger electrode has high quality factors and less modal coupling, which can satisfy the engineering application well. As a conclusion, this study provides a design idea for the electrode to maintain a uniform distribution of quartz crystal microbalance mass sensitivity.

## 1. Introduction

As a typical piezoelectric acoustic resonator operating in thickness shear mode, quartz crystal microbalance (QCM) has the characteristics of simple structure, high mass sensitivity and well measurement resolution, great stability and specificity, and so on. In addition, the development technology of artificial quartz crystal is mature and the manufacturing cost is low nowadays. QCM has greatly contributed to the development of engineering fields such as biology, chemistry, environmental science, food inspections and aerospace during the past decades [1,2,3,4,5].

QCM is usually composed of a specific cut quartz with concentric electrodes on both sides. The commonly used circular electrodes limit the vibration displacement in electrode region to make QCM sensitive to the mechanical and electrical characteristics of the surface load. Moreover, circular electrodes reduce the coupling vibration between fundamental and parasitic overtone modes and increase the reliability of measurement. However, for the traditional QCM with circular electrodes, the mass sensitivity distribution is a Gaussian shape [6], which greatly limits the accuracy and repeatability of precision measurement [7,8,9]. Many scholars have studied the electrode material, structure and thickness to improve such an issue. The early studies on improving the mass sensitivity distribution mainly focused on the analysis of ring electrode and M-N electrode by Josse and Lee [10,11,12]. Such an asymmetric electrode structure can obviously change the distribution of mass sensitivity. Especially, the bi-modal sensitivity distribution of ring electrodes paves a way for exploring the homogenization of sensitivity distribution. The subsequent research mainly focused on optimizing the thickness and geometry of the ring electrode to further unify the mass sensitivity [13,14,15,16]. The bimodal distribution was further modified into three and four peaks through dot ring and dot double-ring by Gao et al. [9], Huang et al. [17]. They compared the influence of electrodes with different sizes on the multi-peak mass sensitivity distribution through theoretical analysis. Aashish et al. [18] and Jiang et al. [19] explored the mass sensitivity distribution of multi-ring electrodes with different proportions based on the finite element method. The investigations have shown that the dot multi-ring electrodes structure can unify the distribution of mass sensitivity. However, less attention is focused on the influence of size and distribution of ring electrodes on the sensitivity. In addition, some of previous electrode shapes are complex and difficult to produce [9], which makes it difficult to apply in practice. Therefore, this paper presents a new type of interdigital electrode as shown in Figure 1, which is similar to the ring electrode but more suitable for engineering application and manufacturing. In this paper, the influences of single finger, dot finger, dot double-finger electrode and the ratio between fingers on the uniformity of mass sensitivity are analyzed through multiple-physical field simulation based on finite element method. According to the summarized design regularity, the mass sensitivity distribution of dot triple-finger electrode with appropriate size ratio is predicted. Finally, an experimental verification of the performance of the presented QCM with dot double-finger electrode is carried out.

## 2. Theory

The distribution of mass sensitivity is a Gaussian when QCM works as a quality sensor. The relationship between the change in resonance frequency and mechanical load is shown as follows [20]:(1)$$\Delta f_{0}=-S_{f}\left(r,\theta \right)\Delta m\left(r,\theta \right)$$
where $S_{f}\left(r,\theta \right)$ is the mass sensitivity function, $\Delta m\left(r,\theta \right)$ is the added mass on surface, r and θ are the position of the added mass in polar coordinates. When QCM works under the fundamental frequency vibration, the mass sensitivity function is as follows [11,17]:(2)$$S_{f}\left(r,\theta \right)=\frac{{\left|\tilde{u}\left(r,\theta \right)\right|}^{2}}{2\pi \int_{0}^{\infty }r{\left|\tilde{u}\left(r,\theta \right)\right|}^{2}dr}C_{f}$$
where $\tilde{u}\left(r,\theta \right)$ is the vibration displacement function of the particle on the surface. The denominator is the integral of the total surface effective particle amplitude intensity. $C_{f}$ is the Sauerbrey sensitivity constant, which is generally used as the average mass sensitivity in practical applications to characterize the linear relationship between the change in resonant frequency and the added mass, the function is shown as follows:(3)$$C_{f}=\frac{f_{0}{}^{2}}{A{\left({\bar{c}}_{66}\rho_{q}\right)}^{1/2}}$$
where $f_{0}$ is the resonant frequency, $\rho_{q}$ is the density of quartz, A is the effective piezoelectric area. ${\bar{c}}_{66}$ is the piezoelectric reinforced elastic coefficient which is the modification of the elastic coefficient $c_{66}$ with the piezoelectric effect. For the AT-cut quartz crystal, the expression is as follows:(4)$${\bar{c}}_{66}=c_{66}+\frac{e_{26}^{2}}{\varepsilon_{22}}$$
where $e_{26}$ is the piezoelectric constant of quartz, $\varepsilon_{22}$ is dielectric constant of quartz.

For a QCM vibrating at the fundamental frequency, the denominator of Equation (2) and the $C_{f}$ are constant. In this case, the mass sensitivity of QCM is proportional to the square of the particle displacement amplitude. Wave propagation is assumed in the $X_{2}$ direction and thickness shear motion is in the $X_{1}$. For AT-cut quartz crystal microbalance, which is operated in thickness shear mode, only the vibration displacement along the $X_{1}$ direction is coupled to the electric potential. Moreover, when QCM vibrates at fundamental frequency, the vibration displacement in $X_{1}$ direction is larger than the displacement in $X_{2}$ and $X_{3}$. The amplitude of vibration displacement in $X_{1}$ has no connection with θ. Therefore, the distribution of mass sensitivity can be reflected by calculating the displacement amplitude distribution in the $X_{1}$ direction of surface particles. The distribution of particle displacement amplitude in cylindrical coordinates can be solved by Bessel equation [9,11,19]:(5)$$r^{2}\frac{\partial^{2}{\tilde{u}}_{r}}{\partial r^{2}}+r\frac{\partial {\tilde{u}}_{r}}{\partial r}+{\left(rk_{r}\right)}^{2}{\tilde{u}}_{r}=0$$
where $k_{r}$ is the wave number of shear horizontal acoustic wave in thickness direction of quartz crystal wafer, the function is shown as follows:(6)$$k_{r}^{2}=k^{2}-k_{i}^{2}=\left(\frac{\pi^{2}}{h^{2}}\right)\left(\frac{f^{2}}{f_{0}^{2}}\right)-\left(\frac{\pi^{2}}{h^{2}}\right)\left(\frac{f_{i}{}^{2}}{f_{0}^{2}}\right),i=E,P,U$$
where k is the wave number of driving frequency, $f_{i}$ and $k_{i}$ are cutoff frequency and corresponding number of cutoff waves in different regions. As shown in Figure 1, E, P, U refers to the fully electroplated region with electrodes on both side of quartz crystal, the partially electroplated region with electrodes on only one side and the non-electroplated region without electrodes on both sides, respectively. h is the thickness of the quartz crystal wafer. The distance between the tip of finger and the other interdigital electrode is narrow. Therefore, the circle region of finger can be equivalent to the fully electroplated region. According to the energy trap effect [21,22,23], when the resonant frequency of QCM satisfies $f_{E}<f_{0}<f_{P}<f_{U}$, waves can spread freely in the fully electroplated region, generating standing wave and forming resonance. In the partial-electroplated region, especially the non-electroplated region, the wave cannot propagate and decays exponentially, therefore, most of vibration energy is confined to the fully electroplated and partial-electroplated region. It is practicable to homogenize distribution of mass sensitivity by changing number, size and position of fingers.

Vibration displacement on surface of QCM with interdigital electrode can be expressed by the solution of Bessel equation [17]:(7)$${\tilde{u}}_{r}=\left\{\begin{array}{c} AJ_{0}\left(k_{r}r\right)+BN_{0}\left(k_{r}r\right){\left(k_{r}r\right)}^{2}>0\\ CI_{0}\left(k_{r}r\right)+DK_{0}\left(k_{r}r\right){\left(k_{r}r\right)}^{2}<\;0\; \end{array}\right.$$
where $J_{0}$ and $N_{0}$ are the Bessel functions of order-zero of the first and second kinds, respectively. $I_{0}$ and $K_{0}$ are the modified Bessel functions of order-zero of the first and second kinds, respectively. A,B,C,D are unknown constant variables, which can be derived by solving linear equations for the continuity condition of displacement and shear strain field at different boundaries [9,11,17]. It should be noted that $N_{0}$ and $K_{0}$ are singular at origin and should be excluded when solving the vibration displacement near the origin. As r goes to infinity, $I_{0}$ tend to infinity. Hence, $I_{0}$ should be excluded when solving the non-electrode region vibration displacement.

## 3. Multiple-Physical Field Simulation

At present, although most analytical and semi-analytical solutions have been proved to be correct near the resonance frequency, many mechanical properties have been ignored due to the simplification of the model. The finite element technique is a powerful numerical tool which has been proved to be able to simulate the thickness shear vibration of quartz crystal microbalances well [24,25,26,27,28,29]. The finite element equation of structural mechanics and electrostatic coupling of AT cut quartz crystal element is as follows [30]:(8)$$\left[\begin{array}{cc} M & 0\\ 0 & 0 \end{array}\right]\left\{\begin{array}{c} \ddot{u}\\ \ddot{v} \end{array}\right\}+\left[\begin{array}{cc} C_{m} & 0\\ 0 & 0 \end{array}\right]\left\{\begin{array}{c} \dot{u}\\ \dot{v} \end{array}\right\}+\left[\begin{array}{cc} K_{m} & K_{me}\\ K_{me}^{T} & K_{e} \end{array}\right]\left\{\begin{array}{c} u\\ v \end{array}\right\}=\left\{\begin{array}{c} F\\ L \end{array}\right\}$$
where M, $C_{m}$ and $K_{m}$ are the mass, damping and stiffness matrices of structure, respectively. $K_{e}$ and $K_{me}$ are the dielectric conduction matrix and piezoelectric coupling matrix are respectively. F, L are the structural force vector and the node charge vector. The dynamic characteristics and vibration displacement of the structure can be obtained based on this equation and the second piezoelectric constitutive equation.

A quartz crystal with the thickness and radius of 0.166 mm and 6.25 mm, respectively, is selected as the substrate to analyze the vibration displacement distribution of single finger, dot finger, dot double finger electrode as shown in Figure 2. There is a circular electrode on back surface with the same radius as the outer contour of interdigital electrode. The orientation of AT-cut quartz crystal is −35.25° based on the IEEE standard on piezoelectricity in 1978. Quartz crystal is an anisotropic piezoelectric material. The material parameter matrix of AT-cut quartz can be obtained from the material constant matrix of the original quartz crystal through coordinated change. The boundary conditions of the quartz crystal are free. A metallic film of gold with a thickness of 100 nm is attached on both of the surfaces of QCM as electrode. When traditional circular electrode radius is 10~20 times of the quartz thickness, the Gaussian distribution can maintain a high peak [31,32]. In order to uniformly distribute the mass sensitivity while maintaining a high amplitude, the radius of bottom circular electrode and the outer contour of top interdigital electrode are always kept at 2.5 mm with the increased number of fingers. An alternating field is applied between the electrodes to keep QCM resonant. The thickness shear motion of quartz crystal can be well-described when there are at least four layers of mesh in the thickness direction. The electrodes are arranged as a grid layer to reduce computing resources. The finite element model of QCM with dot finger electrode is shown in Figure 3. Parameterized simulation is carried out to explore the influence of different proportions between finger and gap width on the vibration displacement.

### 3.1. Single Finger Electrode

The electrode structure of single finger is shown in Figure 2a. In order to analyze the effect of size and position of finger on vibration displacement and obtain the optimal size proportion, five groups of different sizes of finger width $a_{2}$ are selected for simulation, and their widths $a_{2}$ are 2.3, 1.9, 1.5, 0.9 and 0.3 mm, respectively.

Figure 4a shows the normalized displacement distribution in $X_{1}$ direction of quartz surface on electric shaft $X_{1}$ with different $a_{2}$ at the corresponding fundamental resonant frequency. When the finger occupies the main area of single-finger electrode, the displacement distribution starts to appear in double peaks, but the overall distribution is mainly Gaussian shape. As $a_{2}$ decreases, the amplitude of vibration displacement decreases and turns to a more obvious bimodal gradually. This is because the intensity of energy capture decreases with the mass effect of finger electrode decreases. When $a_{2}$ is about 1.5 mm, that is, the proportions of finger width and outer contour radius of the electrode is about 0.6, the displacement distribution is relatively uniform while amplitude can be maintained at a certain level. The vibration displacement nephogram is shown in Figure 4b. It can be seen that the finger is too narrow to capture vibration energy in fully electroplated region when $a_{2}$ is 0.3 mm. In this case, the vibration energy is mainly captured in center by the circular electrode at bottom, which is the main producer of mass effect, and the vibration displacement is a simple convex distribution with a minute amplitude. Although the vibration displacement distribution of the finger electrode is uneven, it provides a design basis for the multi-finger electrodes.

### 3.2. Dot Finger Electrode

A dot electrode is added in the center of partially electroplated region to improve the middle depression of the single finger electrode, its structure is shown in Figure 2b. In order to obtain the influence criteria and the optimal geometric size, the different proportions of dot radius $b_{1}$ and gap width $b_{2}$ are simulated systematically in six groups of different finger and gap sizes as shown in Figure 5.

Parametric simulations are carried out for $b_{1}$ when $b_{2}$ are fixed as 0.2 mm, 0.5 mm, 0.8 mm to analyze the optimum gap width. Figure 5a,b show that when the dot radius is large compared to the gap width, the partially electroplated region cannot improve vibration displacement distribution satisfactorily. Although the vibration displacement presents a convex distribution with small peaks on both sides, the overall vibration displacement still presents an uneven distribution. As the ratio of $b_{1}$ and $b_{2}$ decreases, the vibration amplitude decreases slightly, but the three peaks distribution becomes more obvious. When the ratio of $b_{1}$ and $b_{2}$ is between 0.25 to 0.5 in Figure 5a and 0.5 to 1 in Figure 5b, the displacement distribution is relatively uniform. Figure 5c shows that regardless of the size of $b_{2}$, the vibration displacement presents a convex distribution with small peak on both sides and an extremely high peak in middle. This is because the middle dot electrode dominates the energy distribution due to its large proportion of the electrode area. The simulated results indicate that the displacement distribution is more uniform when the point radius is half of the gap width. Significantly, when the ratio of the dot electrode and the gap width is appropriate, the vibration displacement distribution changes from a bimodal to a convex gradually with the increase in the radius of the dot electrode. This trend indicates that there is an optimal dot electrode radius for the three peaks distribution of mass sensitivity.

A series of parameterized simulations are performed on $b_{1}$ when $b_{2}$ is fixed as 0.4 mm, 0.8 mm and 1.2 mm to study the optimal specific size of the dot electrode. When the gap width increases, the difference between the wave peak and trough will increase gradually, and the distribution become a concave with a low peak in the middle and large peaks on both sides, as shown in $b_{1}:b_{2}=1$ in Figure 5d, $b_{1}:b_{2}=0.5$ in Figure 5e and $b_{1}:b_{2}=0.25$ in Figure 5f. The Figure 5d shows when the ratio of $b_{1}$ to $b_{2}$ is less than 1, that is, the dot electrode radius is less than 0.4 mm, the vibration displacement will not change from multi-peak to extremely uneven convex distribution. When the gap width is 0.8 mm and 1.2 mm, respectively, the appropriate ratio of $b_{1}$ to $b_{2}$ is no more than 1 and 0.5, that is, the dot radius is less than 0.8 mm and 0.6 mm, the non-uniformity convex distribution will not appear even the size of gap width is lager. Meanwhile, the vibration displacements of $b_{1}:b_{2}=1$ in the Figure 5d, $b_{1}:b_{2}=0.5$ in Figure 5e and $b_{1}:b_{2}=0.25$ in Figure 5f confirm that the radius of dot and gap width should be about 0.5 mm to maintain a uniform distribution of vibration displacement. In conclusion, in order to maintain a uniform mass sensitivity distribution of the QCM with dot finger electrode, the radius of the dot electrode should be about 0.5 mm, which accounts for 0.2 of the electrode area, the gap width should be twice the dot electrode radius, and the finger width should be about 0.4 of the electrode area.

### 3.3. Dot Multiple-Finger Electrode

The distribution of vibration displacement of dot finger electrode is further improved by adding a finger on the single finger electrode. The schematic diagram of the dot double-finger electrode structure is shown in Figure 2c. Firstly, in order to study the optimal ratio of the finger width $c_{3}$ and the gap width $c_{2}$, $c_{4}$ of the dot double-finger electrode and its influence on vibration displacement, fixed $c_{2}=c_{4}$. Through the research on dot finger electrode, $c_{1}$ is set at 0.25 mm. The optimum design can be obtained by studying the ratio of the gap width to the inner finger width and the specific width of the finger. Parametric simulations are carried out for $c_{2}$ and $c_{4}$ to investigate the influence of gap width on the overall vibration displacement when the ratio of $c_{3}$ to $c_{1}$ are 1, 1.5 and 2 as shown in Figure 6a–c.

As the finger width exceeds the gap, the vibration displacement starts to shift to multiple peaks distribution, but it still presents a Gaussian distribution; the partially electroplated region has little effect. When the ratio of the finger width to the dot radius increases, the uniformity of vibration displacement distribution becomes worse when the size proportion is appropriate. Figure 6a shows that the displacement distribution changes from Gauss to multimodal gradually with the increase in the gap width. Especially, when finger width is 0.5 to 1 times of gap, the distribution is uniform. The optimum ratio in Figure 6b,c are between 0.75 to 1.01 and 1 to 2, respectively. The simulated results indicate that with the increase in the ratio of $c_{3}$ to $c_{1}$, the gap width which can maintain a uniform distribution decreases gradually. In this case, no matter what the ratio of gap to finger width is, the displacement distribution is uniform only when the ratio of the outermost finger width $c_{5}$ to the total electrode radius is about 0.4. It is similar to the ratio of the finger to the total electrode radius in the dot finger electrode. It illustrates that the dot double-finger electrode improves displacement distribution on the basis of the dot finger electrode. The concave area of distribution is compensated by the added finger rather than distributing the ratio of the partially electroplated region evenly to unify the distribution of vibration displacement.

$c_{2}:c_{1}$ is set as 1, 1.5, and 2, respectively, to study the influence of finger size and position on displacement distribution with different $c_{3}$. With the increase in $c_{3}:c_{2}$, the displacement distribution starts to change from three peaks to uniform multi-peak distribution gradually when the gap width $c_{2}$ is 0.25 mm. When the effect of the partially electroplated region becomes more intense, the distribution of displacement becomes a multimodal jittered distribution with lager difference between peak and trough. Finally, the distribution of displacement becomes a convex distribution due to the large ratio of the finger width to the partially electroplated region. The appropriate ratio of $c_{3}$ and $c_{2}$ is between 0.48 and 1. Figure 6d,e show that with the increase in $c_{1}$, the displacement distribution has become a more uniform five peaks distribution compared with the original three peaks distribution when $c_{3}:c_{2}$ is about 0.2 and become a jittered distribution compared with the original convex distribution when $c_{3}:c_{2}$ is about 3. The partially electroplated region plays a role in balancing the excessive mass sensitivity of the whole electrode region when the gap width is 0.375 mm, and the appropriate ratio of finger and gap width is between 0.51 and 0.986. The displacement distributions shown in Figure 6f are improved compared to the same ratio of $c_{3}$ and $c_{2}$ in Figure 6d,e, which indicates that the large gap width can improve the distribution of mass sensitivity greatly. In this case, the optimal ratio of finger and gap is 0.5.

The dot double-finger electrode when a relatively uniform vibration displacement distribution, which size is $c_{1}=c_{3}=0.25\;\mathrm{mm}$, $c_{2}=c_{4}=0.5\;\mathrm{mm}$, is selected for further research. The ratio of $c_{2}$ and $c_{4}$ are changed to explore the effect of different ratio between the gap width on mass sensitivity. The normalized vibration displacement with different ratio of $c_{2}$ and $c_{4}$ are shown in Figure 7. With the increase in $c_{2}:c_{4}$, the amplitude of displacement increases slightly. The distribution of displacement becomes more uniform, and the difference between peak and trough decreases. However, the displacement distribution will be concave when the ratio of $c_{2}$ and $c_{4}$ is large. The decrease order gap width from central to edge can improve the amplitude of central displacement and make the peak distribute at the center as much as possible in order to maintain a uniform mass sensitivity distribution. In summary, the design criteria of the dot double-finger electrode are the same as that of the dot finger electrode. The width of the outer finger should be maintained at about 0.4 times of the electrode area, and the gap width should be twice of finger width. It should be noted that the width of the finger should be kept smaller than the dot electrode radius, and the gap should obey the decrease order from the center to the periphery to maintain the uniform distribution of mass sensitivity when design the dot multi-finger electrode.

According to the summarized design criteria, the vibration displacement in the $X_{1}$ direction of the dot triple-finger electrode, which is shown in Figure 1, is obtained. The size of the inner finger and the gap width are 0.15 mm and 0.35 mm, respectively. The radius of the middle dot electrode and the width of the outer finger are 0.15 mm and 1 mm. Figure 8 shows the vibration displacement distribution comparison of original circular electrode and dot triple-finger electrode. The results show that the vibration displacement of QCM with the new interdigital electrode rises faster at the edge of the electrode compared with the circular electrode. Compared with the circular electrode, the dot triple-finger electrode can make sensitivity distribution more uniform while maintaining the amplitude at a certain level. The displacement distribution of dot triple-finger electrode illustrates that the design criteria are correct and universal. It can be extended to design interdigital electrodes with more fingers.

## 4. Experiment

Interdigital electrodes with a thickness of 100 nm were fabricated by evaporation coating on AT-cut quartz crystals with a cut angle of 35.25. A series of QCM with dot double-finger electrode were prepared, and their performance were experimentally tested. The physical structure is shown in Figure 9. The electrode material is gold, and the specific dimensions of the quartz crystal and electrode are shown in Table 1.

The performance parameters of QCM were tested by energizing interdigital electrodes and bottom circular electrode with an impedance analyzer and probe platform simultaneously, as shown in Figure 10. In order to maintain a well contact between the measuring probe and the electrode, the QCM was measured on the stage of an industrial microscope. The bottom circular electrode and the top interdigital electrode are in close contact with to the two probe bases, respectively, through a conductive gel on the objective table. The QCM was measured by the network analyzer in range of 9.75 MHz~10.25 MHz with an accuracy of 0.0025 MHz. All tests were performed in a clean room at room temperature. The results are shown in Table 2.

The equivalent dynamic resistance $R_{1}$ represents the energy loss of the quartz crystal resonator and reflects the vibration damping of resonator. The static capacitance $c_{0}$ is the additional capacitance caused by the gold electrode plated on both sides of quartz crystal and represents the electrostatic coupling between the two electrodes. The quality factor Q is the ratio of the stored peak energy to the energy consumed in each cycle, which is a measure of the energy loss in the resonator and reflects the overall performance of the resonator. A high Q value and a low $c_{0}$ value usually represent a high-performance resonator. The dynamic resistance of a 6 MHz QCM with circular electrodes is generally about 50 Ω, while the dynamic resistance of the prepared QCM is about 4520 Ω. The reason for the high resistance may be that the width of lead electrode, which is in contact with the probes, is undersized (0.2 mm). The resistance measured is inaccurate due to poor contact between probes and electrodes. The material of probes is tungsten, and a conductive gel is used to keep the probes in contact with the bottom and top surface electrodes which cause excessive contact resistance. Such resonators may stop vibration when operating in some special environments, especially in liquid environments. The admittance of three groups of QCM with significant differences in quality factors were studied, as shown in Figure 11. The disordered degree of the crest of the admittance curve can reflect the degree of modal coupling. The admittance wave peak of group three and nine are sharp. The peaks of the stray mode are small, and the disturbance between the stray mode and the fundamental mode are mild. The close distance between the stray mode and the fundamental mode will aggravate the modal coupling problem. The distribution of admittance peaks of the group one is disordered, and the modal coupling problem is serious. In a word, although the resistance of the prepared QCM is large and it may stop vibration in some specific environments, the quality factors are all around 53,973, and the mode coupling level can be accepted. The resonator, which is manufactured with the design criteria of uniform mass sensitivity, shows a satisfactory performance and can be well-applied in the manufacturing of hydrogen atmosphere sensors and other practical engineering tools in the future.

## 5. Conclusions

In this paper, the design criteria are summarized by analyzing sensitive structural parameters of the interdigital electrode. The theoretical calculation and simulation show that the design of the three types of electrodes is the same, instead of redistributing the size and position of fingers and gaps evenly to unify the distribution of mass sensitivity. A large dot electrode will cause the vibration displacement to be a pronounced convexity distribution. When the finger width is half the gap, the mass sensitivity can be distributed more evenly. The average quality factor of the QCMs with dot double-finger electrode, which are manufactured with the criteria, is 53,973. The performance and noise level of admittance can satisfy the engineering application well. The interdigital electrode proposed in this paper is expected to provide design reference for the engineering application of QCM with uniform mass sensitivity distribution.

## Figures and Tables

**Figure 1 sensors-22-07422-f001:**
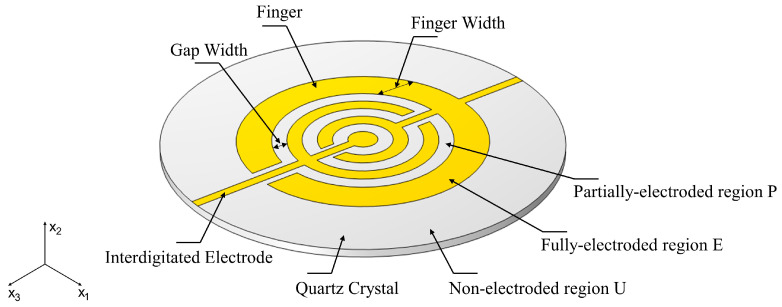
Structure of QCM with ring-shaped interdigital electrode.

**Figure 2 sensors-22-07422-f002:**
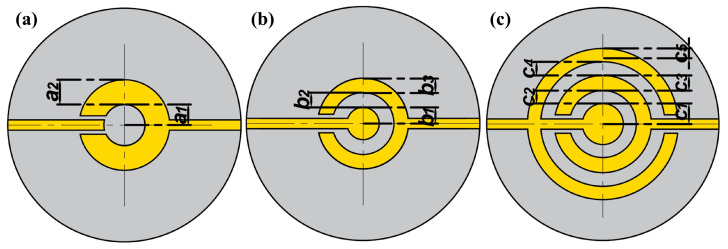
Structure of interdigital electrode. (**a**) Single finger electrode. (**b**) Dot finger electrode. (**c**) Dot double-finger electrode.

**Figure 3 sensors-22-07422-f003:**
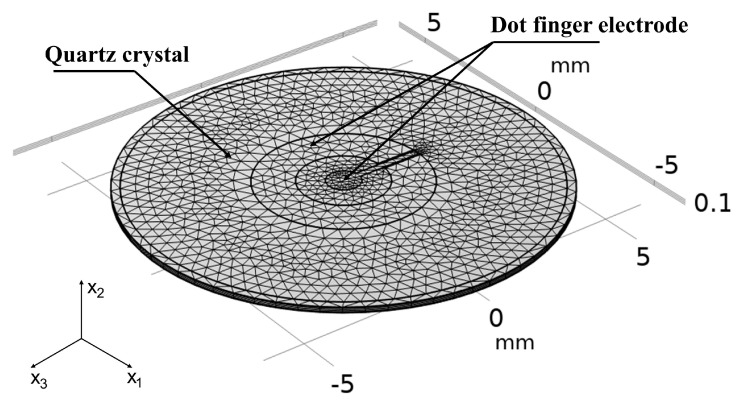
The finite element model of QCM with dot finger electrode.

**Figure 4 sensors-22-07422-f004:**
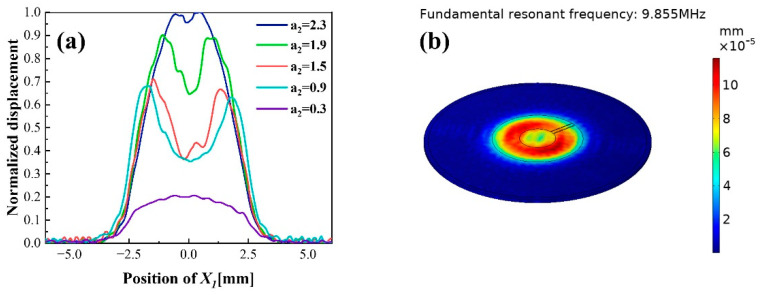
The vibration displacement in $X_{1}$ direction of QCM with single finger electrode: (**a**) Normalized displacement distribution on electric shaft $X_{1}$ with different finger width $a_{2}\;$ (mm); (**b**) Vibration displacement nephogram when $a_{2}$ is 1.5 mm.

**Figure 5 sensors-22-07422-f005:**
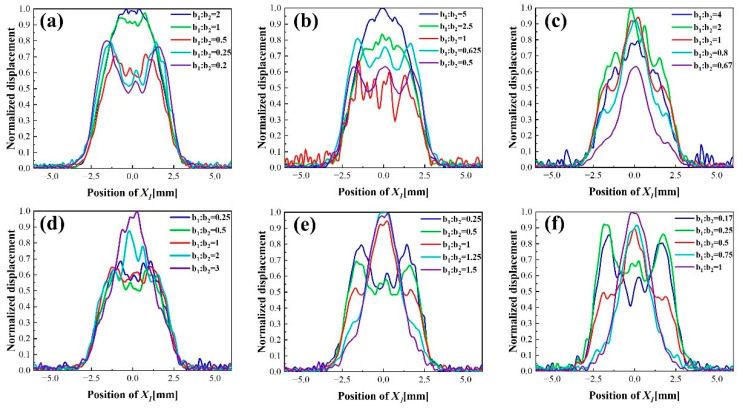
Normalized displacement distribution in $X_{1}$ direction on electric shaft $X_{1}$ of the dot-finger electrode: (**a**–**c**) Fixed dot electrode radius $b_{1}$ is 0.2 mm, 0.5 mm, 0.8 mm with different ratio of $b_{1}$ to $b_{2}$; (**d**–**f**) Fixed gap width $b_{2}$ is 0.4 mm, 0.8 mm, 1.2 mm with different ratio of $b_{1}$ to $b_{2}$.

**Figure 6 sensors-22-07422-f006:**
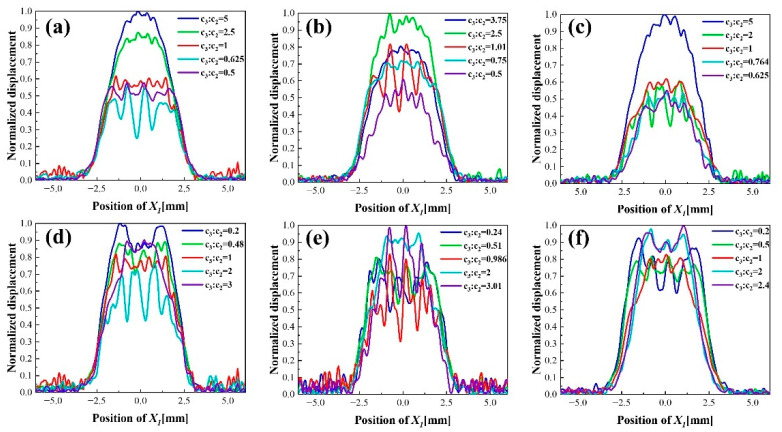
Normalized displacement distribution in $X_{1}$ direction on electric shaft $X_{1}$ of dot double-finger electrode: (**a**–**c**) Fixed dot electrode radius $c_{3}$ is 0.25 mm, 0.375 mm, 0.5 mm with different ratio of $c_{3}$ to $c_{2}$; (**d**–**f**) Fixed gap width $c_{2}$ is 0.25 mm, 0.375 mm, 0.5 mm with different ratio of $c_{3}$ to $c_{2}$.

**Figure 7 sensors-22-07422-f007:**
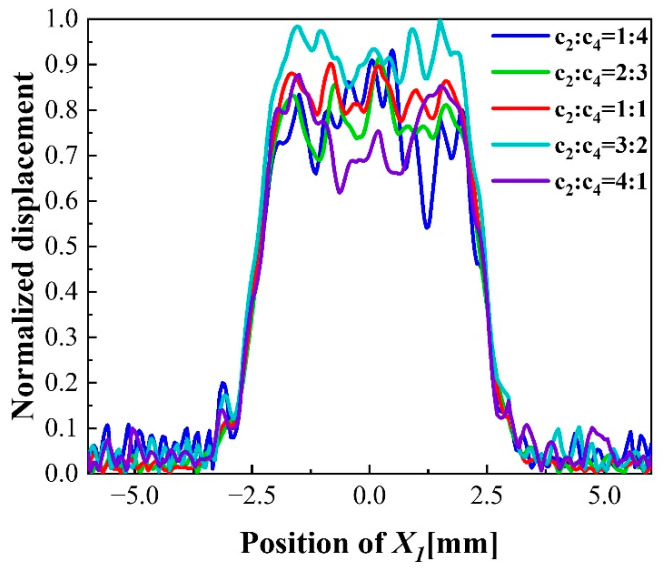
Normalized displacement distribution with different ratio of $c_{2}$ and $c_{4}$.

**Figure 8 sensors-22-07422-f008:**
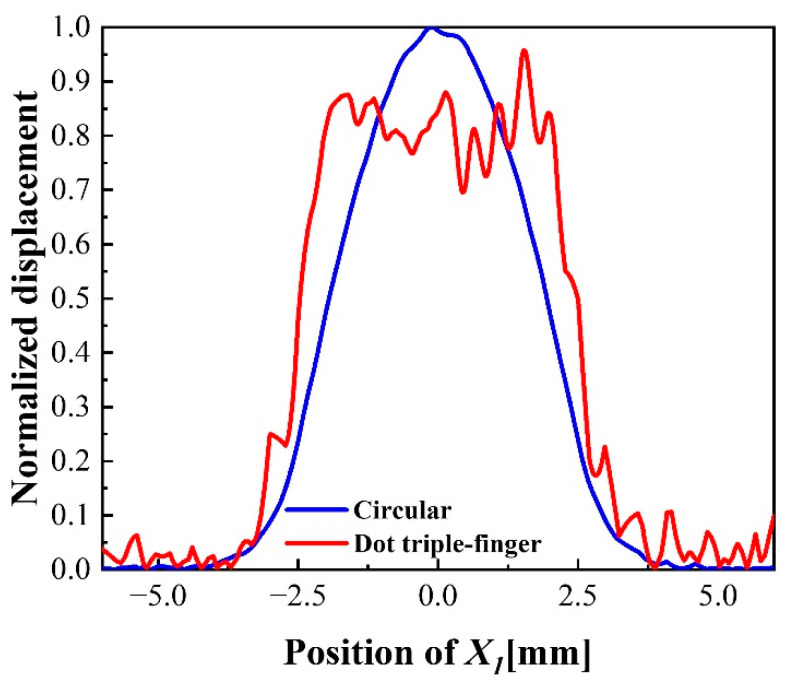
Vibration displacement distribution of dot triple-finger and circular electrode.

**Figure 9 sensors-22-07422-f009:**
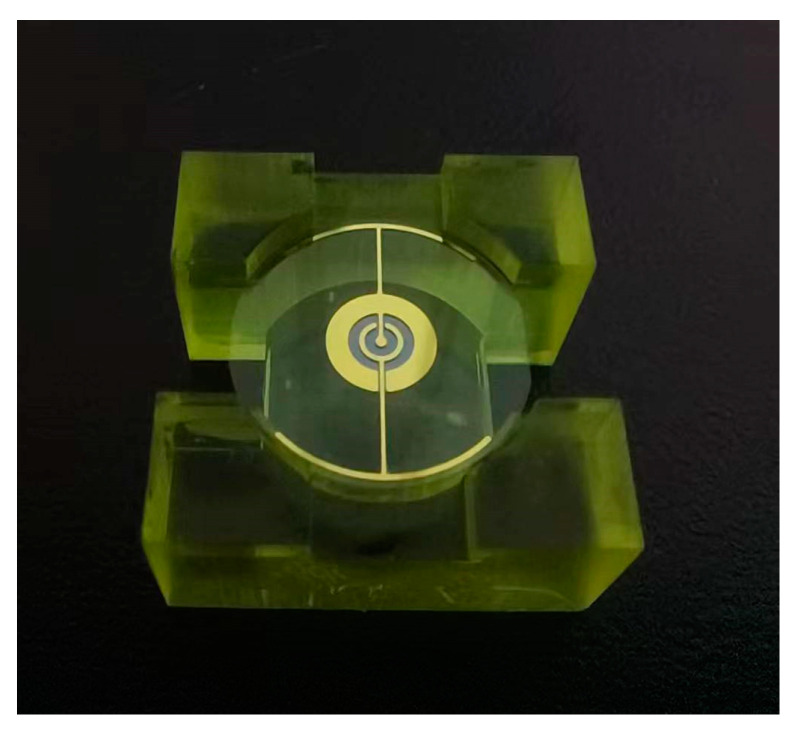
Photograph of the fabricated the QCM with double-finger electrode.

**Figure 10 sensors-22-07422-f010:**
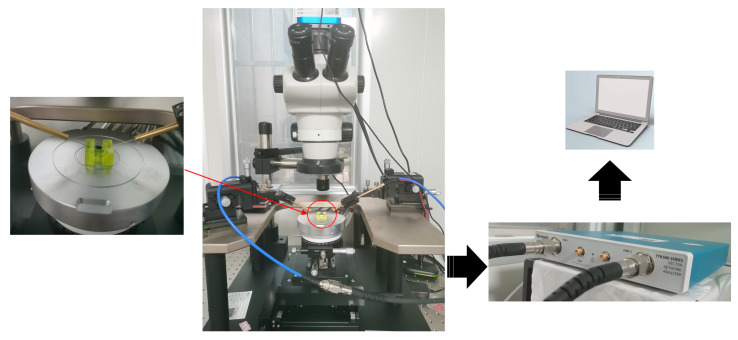
The experiment flow chart of the QCM with interdigital electrode.

**Figure 11 sensors-22-07422-f011:**
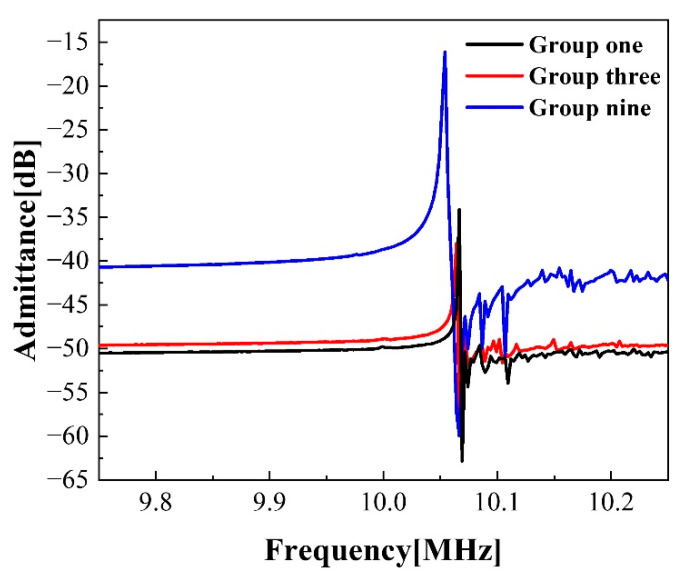
Admittance characteristics of the QCM with double-finger electrode.

**Table 1 sensors-22-07422-t001:** Dimensions of quartz crystal and double-finger electrode.

Object	Parameters	Size (mm)
Quartz crystal plate	Thickness	0.17
	diameter	13.69
Interdigital electrode	$\mathrm{Dot}\;\mathrm{radius}\;c_{1}$	0.25
	$\mathrm{Gap}\;\mathrm{width}\;c_{1},\;c_{4}$	0.5
	$\mathrm{Finger}\;\mathrm{width}\;c_{3}$	0.25
	$\mathrm{Outer}\;\mathrm{finger}\;c_{5}$	1

**Table 2 sensors-22-07422-t002:** Measurement characteristics of QCM with double-finger electrode.

Serial Number	Resonant Frequency (MHz)	Quality Factor	Dynamic Resistance (Ω)	Static Capacitance (pF)
1	10.068	57,710	2686	1.13
2	10.068	53,140	5372	1.05
3	10.066	78,680	3239	1.21
4	10.063	43,640	5006	1.18
5	10.067	74,160	3054	1.17
6	10.067	43,420	6394	1.14
7	10.067	46,690	4965	1.08
8	10.068	46,950	5859	1.11
9	10.065	41,370	4108	1.07
Average	10.067	53,973	4520	1.13

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
