# Peer review of "Design and Analysis of Quartz Crystal Microbalance with a New Ring-Shaped Interdigital Electrode"

_sensors, 2022, doi:10.3390/s22197422_

Round 1
Reviewer 1 Report
Review of: Design and Analysis of Quartz Crystal Microbalance with a New Ring-like Interdigital Electrode
By: Pengyi Wang et al.
The paper deals with the concept proposal, design, fabrication and test of a ring-like interdigital electrode to improve the accuracy and repeatability of quartz crystal microbalances. The subject is suitable for the journal Sensors. The work is clearly explained. My main concern is related to the poor description of the experimental part and the complete lack of experimental validation, as explained below. These issues should be solved in order to meet the quality standard of the journal Sensors.
Figure 1. Back surface is not electroded, right?
Equation 2. Please define subindex “1” Does it refer to direction “1” in Figure 1?
Equation 3. Please explain what doe you mean with “enhanced sehar modulus and density of quartz”
Line 85. Why X1? Are axis shown in figure 1 correct?
Section 3. Some additional information about the FEM modelling would be good. Package used (commercial or developed by the group), some comments on the mesh and computational resources required. Time to solve one particular configuration, etc…
Line 126. It may be good to mention the orientation of the quartz crystal cut.
Figure 2. Do you have then a back electrode? As the radius of the outer contour top electrode is 2.5 mm. Does it mean that size is the same and as you increase the number of rings, width and diameter of the rings become smaller?
The comparison of the performance of the different proposed configurations is based on the visual analysis of the both peak amplitude and “smoothness” of the distribution in the graph of normalized displacement vs position. It would we very interesting if authors can define a figure of merit to quantify the goodness of a given design. This will permit a more quantitative analysis and also the use of complex optimization techniques to find out the best electrode configuration.
Figures 6 and 7. The dot-double finger configuration results are shown in figure 6 and the dot-triple finger in figure 7 (check data label in figure 7 !), where the result of the circular electrode is also show, In general, these optimized configuration for the n-finger electrode present a flatter response in the full width of the electrode area, but also a certain level of noise in the central “plateau”, where normalized amplitude can oscillate between 0.7 and 0.95. The main question is if this “noise” is acceptable for these sensors and if the gain in the sensor response obtained by the wider distribution (compared with circular electrode) is larger than the “loss” produced by the amplitude decrease and this “noise”.
Authors mention that for the experimental analysis the double finger configuration is selected. What is the reason for this? Is not the triple finger better of more flexible, from the design point of view?
Experiment.
Please explain or give more information related to:
Quartz: type of cut, and manufacturer
Electrode: thickness, technique and equipment used for the deposition (manufacturer and model). Please mention how thickness and shape is controlled.
Table 2. Please, explain equipment used for these measurements: manufacturer and model and the configuration used. It is very important that you include the error in the measurements.
Line 287. You mention figure 7, but this is figure 8.
Line 307. You mention figure 8, but this is figure 9.
Figure 9. Better change units in x-axis to MHz
I miss some comparison between measured response and calculated one. I find no reason why you don’t include a comparison of the measured and calculated admittance (for example).
Moreover, all design work has been performed based on the comparison between the distribution of the vibration amplitude. I see no reason why you haven’t included a comparison, at least for one case, of the measured and the calculated vibration amplitude distributions.
No example of the use of the sensor is proposed. It is claimed that this configuration can improve sensor sensitivity, but it will be good to see one application example.
Conclusions.
Authors mention: “In conclusion, the size and position of gap and fingers indeed have a significant effect on the distribution of mass sensitivity.”
However this is purely theoretical, to really prove that, authors should complete the experimental analysis as suggested above.
Reviewer 2 Report
In this work the authors present a new electrode design for QCM that could provide a more uniform, less Gaussian, mass sensitivity across the measuring surface. The authors take a systematic approach to the design, considering single-finger, dot finger, and dot double-finger formats. Rigorous computational methods were used to estimate the sensitivity distribution in each format and with a range of dimensions. This led the authors to the fabrication and testing of one specific design. The theory of QCM and issues with current electrode designs is well described. Computational models are well described, with a few suggestions for improvement noted below. For fabrication and testing, a methods section is lacking and drawbacks or limitations of their design are not discussed. These results should be of interest and use to the acoustic wave sensor community, but a number improvements should be made to the manuscript.
1. In general, writing style should improve to properly convey each point with grammatical correctness.
2. Figure 3 – The term “beta” should be better described since it is used prominently in discussing these results. In general, the authors could more clearly explain what profile would be ideal in a plot like Figure 3(a).
3. Figure 6 – The data presented in Figure 6 and in other prior figures is based on a computational algorithm, which does not really have statistical significance. Asking if these traces are statistically different from each other is not really relevant, but are they practically different from each other? How is one set of dimensions selected to move forward?
4. Experimental Section – No experimental details are provided in this section. How were these devices fabricated? Were the measurements made in Table 2 collected in air, vacuum, water? The data in Table 2 is not well analyzed. What is the average and standard deviation of Quality Factors? Is this acceptable? Why does QCM 1 have a resistance of 408 Ohms when the rest are in the thousands? Report the average and standard deviation of these.
5. Is the very high resistance problematic with these QCM crystals. In my experience, such high resistance values begin to cause problems in QCM measurements.
6. Quality factors are referred to as being around 50000 in the text, but are reported as being around 50 in the Table. Which is correct?
7. How does this electrode configuration influence surface functionality with affinity ligands? If an antibody were to be conjugated to the surface of this sensor via a self-assembled monolayer, the antibody would not be uniformly distributed. Is this a limitation of this design?
Round 2
Reviewer 1 Report
Paper is acceptable in its present for.
Authors produced a detailed response letter. But not all comments included in the response letter are included in the paper. Authors should consider the addition of this information ans readers may have the same questions I had.